

# Dynamic Mode Decomposition of Extreme Events

Maša Ann[1], Jörn Behrens[2], and Jana Sillmann[3]

[1]Universität Hamburg
[2]Universität Hamburg
[3]Universität Hamburg

**Correspondence:** Maša Ann (masa.ann@uni-hamburg.de)

**Abstract.** Most data-driven methods, among them Dynamic Mode Decomposition (DMD), focus on analysing and reconstructing the average behavior of a system. However, the primary interest often lies in the anomalous behaviour, known as extreme events. This is especially the case in climate research, where extreme events have significant economic and societal costs. Therefore, we extend a DMD method to account for extreme events by adding a penalisation term. This extension allows us to not only better reconstruct the extreme events, but also extract the spatio-temporal structures related to those extreme events. DMD was originally developed by Schmid and Sesterhenn (Schmid and Sesterhenn, 2008) to enable the fluid dynamics community to identify spatio-temporal coherent structures (called *modes*) from high-dimensional data. In its essence DMD uses most relevant modes to filter the noise and reconstruct the original signal. We ask "Is the noise really noise"? Or can we attribute some of these dynamic modes, that result from the DMD, to extreme events? We applied this new method to the climate system, well known for its high-dimensionality. We examined two heatwaves that occurred in Europe (HW 2003 and HW 2010). In both cases we were able to improve the accuracy of the reconstruction. This novel variation of the DMD, can also be applied to other dynamical systems across many disciplines, in which extreme events are of interest.

## 1 Introduction

Dynamic Mode Decomposition (DMD) is a relatively recent algorithm introduced by Schmid and Sesterhenn (Schmid and Sesterhenn, 2008). Originally developed for modeling fluids, DMD quickly developed as a powerful tool for analyzing the dynamics of nonlinear systems (H. Tu et al., 2014). It has also found its applications across many different disciplines (Brunton et al., 2021), e.g. neuroscience (Brunton et al., 2016), epidemiology (Proctor and Eckhoff, 2015), fluids mechanics (Schmid, 2010; Rowley et al., 2009b), video analysis (Erichson et al., 2016), climate dynamics (Froyland et al., 2021; Chowdary et al., 2023).

The foundational concept of DMD is rooted in the well-established tradition of signal decomposition. DMD is a dimensionality reduction technique that transforms data from a high-dimensional space to a low-dimensional one while preserving the essential features of the original data. The resulting reconstruction typically captures the system's average behavior effectively but often falls short in representing anomalous or extreme events (Lucarini et al., 2016).

DMD was initially proposed as a dimensionality reduction technique that extracts dominant spatio-temporal coherent structures from high-dimensional time-series data. Motivated by the Occam's razor principle - *The simplest explanation is usually*




*the best one* - our approach tries to extract a single most significant modes that could enlighten the spatial pattern of extreme events. The trade-off between model complexity and accuracy is already explored in (Jovanović et al., 2014). However, the focus there is solely on the sparsity of the model and does not address the reconstruction of extreme values. DMD has significant potential to improve our understanding of the way in which coherent structures in the atmospheric flows evolve and interact (Duke et al., 2012). We frame the problem as an optimization task that places greater emphasis on the extreme events.

DMD is a practical implementation of the *Koopman theory* in the context of data-driven analysis. Koopman theory focuses on the *Koopman operator*, an infinite-dimensional linear operator that evolves observables over time. It provides a way to analyse the behaviour of nonlinear systems using linear methods - a powerful tool for understanding complex dynamics (Koopman, 1931; Gaspard, 1998).

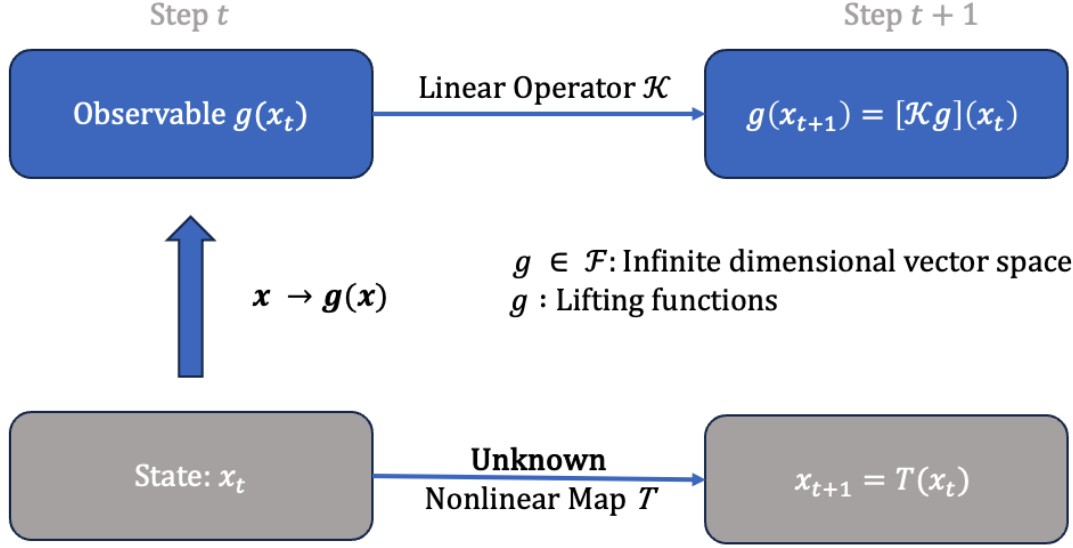

**Figure 1.** Sketch of the Koopman Operator Theory. Adapted from Shi et al. (2024)

**Definition 1.** *The **Koopman operator** is a linear operator that describes the evolution of observables in a dynamical system. Let $\mathbf{x}_{t+1} = f(\mathbf{x}_t)$ define a discrete-time dynamical system, where $\mathbf{x}_t \in \mathbb{R}^n$ represents the state at time $t$ and $f$ is the state transition map.*

*An **observable** is a function $g : \mathbb{R}^n \to \mathbb{R}$ that measures some property of the system state. The Koopman operator, $\mathcal{K}$, acts on*
*the observable $g$ to describe its evolution:*

$$(\mathcal{K}g)(\mathbf{x}) = g(f(\mathbf{x})),$$

*or equivalently, in terms of time:*

$$g(\mathbf{x}_{t+1}) = (\mathcal{K}g)(\mathbf{x}_t).$$



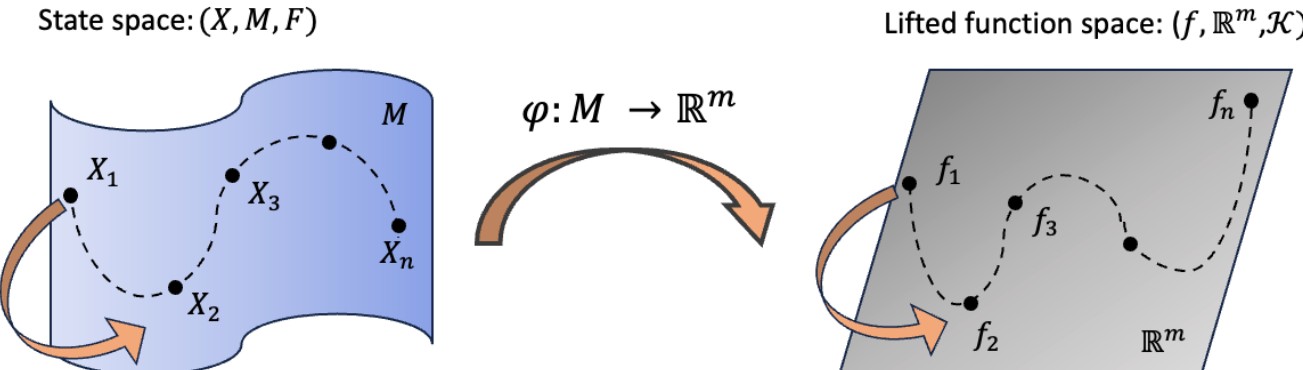

**Figure 2.** Figure adapted from Wang et al. (2023), demonstrating uplifting.

In other words, the Koopman operator is a linear operator that governs the evolution of scalar functions (often referred to as observables) along trajectories of a given nonlinear dynamical system. Each of the individual measurements may be expanded in terms of the eigenfunction $\varphi_j(\mathbf{x})$, which provide a basis for a Hilbert space

$$g(\mathbf{x}_t) = \sum_{j=1}^{\infty} v_{ij}\varphi_j(\mathbf{x}) \tag{1}$$

where $\mathbf{v}_j = (v_{1j}, \ldots, v_{lj}, \ldots)^{\top}$ is the $j$th *Koopman mode* associated with the eigenfunction $\varphi_j$. These Koopman modes are coherent *spatial* modes that behave linearly with the same temporal dynamics and are known as dynamic modes in DMD. Now, it is possible to represent the dynamics of the measurements $g$ as follows:

$$g(\mathbf{x_t}) = \mathcal{K}_{\Delta t}^t g(\mathbf{x_0}) = \mathcal{K}_{\Delta t}^t \sum_{j=1}^{\infty} v_{ij}\varphi_j(\mathbf{x_0}) = \sum_{j=1}^{\infty} \mathcal{K}_{\Delta t}^t v_{ij}\varphi_j(\mathbf{x_0}) = \sum_{j=1}^{\infty} \lambda_j^t v_{ij}\varphi_j(\mathbf{x_0}) \tag{2}$$

where $\mathcal{K}_{\Delta t}^t$ is Koopman operator applied $t$ times. The triple $(\lambda_j, \varphi_j, \mathbf{v_j})_{j=0}^{\infty}$ is known as *Koopman mode decomposition* (Mezić, 2005).

The connection between Koopman mode decomposition and Dynamic Mode Decomposition (DMD) was established in (Rowley et al., 2009b). DMD is an algorithm for approximating the Koopman operator (Mezić, 2013; Rowley et al., 2009a), yielding a triple: (i) DMD eigenvalues approximate Koopman eigenvalues $\lambda_j$, (ii) DMD modes approximate Koopman modes $\mathbf{v}_j$ (denoted $\phi_j$ in DMD), and (iii) DMD mode amplitudes approximate Koopman eigenfunctions evaluated at the initial condition, $\varphi_j(\mathbf{x_0})$ (denoted $b_j$ in DMD).

The main research question is: Can we find the spatial modes that are responsible for an extreme event?

When modeling dynamical systems, dimensionality reduction algorithms, as their name indicates, reduce the dimensionality, i.e. the complexity of the system replacing it with simpler models. The common challenge when reconstructing (1), is the



question how many modes are needed. This is the balancing act between the simplicity and correctness of the model. By adding less modes we get the signal that is a *smoothed* reconstruction compared to the original signal. By *smooth* we mean averaging the signal over time, omitting the extremes. Adding more modes improves the reconstruction (e.g., reducing the mean squared error), but it also makes the reconstruction less smooth and causes the extreme values to become more pronounced. In other words, adding more modes we approximate extremes better.

The questions arise - Can we identify those modes that are responsible for better reconstruction of the extremes? If yes, can we physically interpret these coherent spatial patterns called modes? In spectral analysis, the modes with the highest energy are typically considered. But do the modes with lower energy play a role in extreme events, or are they rightfully dismissed as mere noise?

Probably the most *famous* and widely used in the dimensionality reduction of physical systems is the Proper Orthogonal Decomposition (POD), also known by other names such as Principal Components Analysis (PCA), the Karhunen-Loeve (KL) decomposition, or empirical orthogonal functions (EOF)(Lorenz, 1956). In climate science EOFs are essentially the result of applying PCA to meteorological data. Consequently, EOF analysis is mathematically equivalent to PCA, and by extension, to POD and Singular Value Decomposition (SVD) (H. Tu et al., 2014). In many nonlinear dynamical systems, snapshot data (measurements) often reveal low-dimensional characteristics, where most of the variance or energy is concentrated in a few modes obtained through SVD. The modes are spatial fields that often identify coherent structures in the flow. However, to the best of our knowledge, all these methods focus on the average behaviour of the nonlinear dynamical system at hand.

Whether we are talking about the numerical method of PCA in statistics (Hotelling, 1933) or POD in fluid mechanics (Berkooz et al., 1993), they both rely on SVD, where the decomposition results are orthogonal vectors that are used for the reconstruction of the original signal. However, the assumption of orthogonality is omitted in the DMD, but the rules of reconstruction stay the same. By omitting the orthogonality, the DMD modes can also be more physically meaningful.

Originally, DMD has been derived from POD (Holmes, 2012; Schmid, 2010; Schmid et al., 2012). However, as already mentioned, POD modes are orthogonal, DMD are not. Additionally, DMD modes are dynamically invariant, POD are not. While POD is based entirely on spatial correlation and energy, DMD adds the temporal information as well. DMD performs a modal decomposition where each mode consists of spatially correlated structures that can be related to certain oscillatory evolution in time. This is a result of the fact that POD solves the eigenvalue problem for the covariance matrix of the data, while DMD employs a time-shifted cross-correlation matrix capturing the linear dependence of the snapshots at the next time step on the snapshots at the current time step (Rowley et al., 2009b; Zhang et al., 2014; Smith et al., 2005).

One of the primary assumptions is that only a few key terms in equation (1) govern the dynamics of a system, resulting in sparse equations within the extensive space of potential functions. Therefore we apply the sparse regression to identify the minimal and most effective terms that accurately capture the underlying dynamics of observables. The resulting model is then parsimonious, striking a balance between complexity and descriptive power, while avoiding overfitting. A similar approach is taken in (Jovanović et al., 2014) where the extension of DMD is developed also with a focus on sparsity of the modes. However, unlike in our extension, the focus there is not on the extremes, but rather on the average signal behaviour.





Our method holds particular promise for addressing extreme events in the climate system, offering substantial potential for applications across engineering, the physical sciences, and the biological sciences. In climate science, DMD can appear under the name of Linear Inverse Modelling (LIM). Although the algorithms differ, they become equivalent under these three conditions - i) the LIM is performed in the space of EOF coefficients (rather than in physical space), ii) the EOFs are computed from snapshots alone (no external forcing) (H. Tu et al., 2014) and iii) LIM assumes no stochastic noise.

## 2  Methodology

For better understanding we provide the following illustration of the DMD decomposition (Fig. 3). The input to the method consists of snapshots, with each snapshot representing the atmospheric state at a specific time step in our experiments. The resulting triplet of the DMD mentioned before (modes, amplitudes, evolution) is well depicted here.

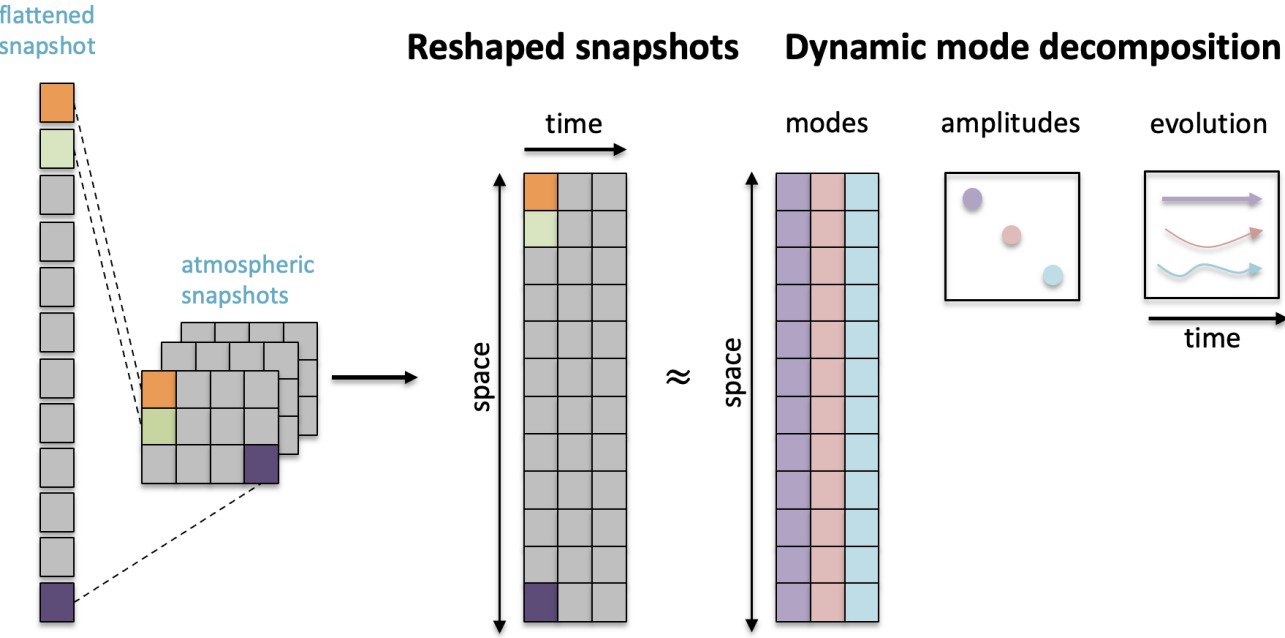

**Figure 3.** Illustration of the DMD decomposition adapted from Erichson et al., J. Real-Time Image Process., 2016.

In this work we will focus on the **amplitudes** matrix and its role in explaining extreme events. The **modes** represent distinct spatial patterns that grow or decay with different frequency over time. The information about growth and decay is captured with the matrix **evolution**. Modes are ordered by their frequencies, with the first modes representing the dominant behaviors (Smith et al., 2005; Rowley et al., 2009b; Zhang et al., 2014). Each mode has a corresponding amplitude, that holds the information about the energy, i.e. the overall importance for the reconstruction.



## 2.1 Discretisation

In theory, we are interested in the dynamical system and its evolution:

$$\frac{d\mathbf{x}}{dt} = f(\mathbf{x}, t; \mu) \tag{3}$$

with a high-dimensional state $\mathbf{x}(t) \in \mathbb{R}^m$. The dynamics $f(\mathbf{x}, t; \mu)$ is unknown, only the observations of the system state can be used to approximate the dynamics and predict the future state. In practice, we are analysing its discrete-time flow map

$$\mathbf{x_{k+1}} = F(\mathbf{x_k}) \tag{4}$$

We search for the operator $A$ that linearises the dynamics in form of:

$$\frac{d\mathbf{x}}{dt} = A\mathbf{x} \tag{5}$$

that has a closed-form solution:

$$\mathbf{x}(t_0 + t) = e^{At}\mathbf{x}(t_0) \tag{6}$$

The solution to this system may be expressed simply in terms of eigenvalues $\lambda_k$ and eigenvectors $\phi_k$ of this discrete map $A$.

$$x_{k+1} = \sum_{j=1}^{K} \phi_j \lambda_j^{k+1} b_j \tag{7}$$

where $b_j \in \mathbb{C}$ , also called amplitude, contains the weights of the modes, the eigenvalues $\lambda_j \in \mathbb{C}$ represent the temporal be-
125 haviour of the system and the eigenvectors $\phi_j \mathbb{C}^n$ are the spatial modes. Equation (7) is a mathematical formulation of Fig. 3.

Depending on the dynamics, we might be able to perform such a decomposition exactly using finitely many modes; in real and realistic flows, this is highly unlikely. Instead, we aim to achieve the decomposition only approximately, by retaining a relatively small number of modes $K$.

## 2.2 Regression

The DMD method produces a low-rank decomposition of the matrix $A$ by optimising the fit to the trajectory $x_k$

$$\|x_{k+1} - \mathbf{A}x_k\|_2 \tag{8}$$

where $x_k$ and $x_{k+1}$ represent measurements taken at two consecutive timestamps, separated by time interval $\Delta t$.

This linearisation holds only locally as we are finding optimal local linear approximations of snapshots, where napshots are
135 states of the system sampled in time.

Since we want to minimise the error across all snapshots, we reshape all $m$ snaphots into a high-dimensional column vectors and arrange them in two large matrices:





$$\mathbf{X} = \begin{bmatrix} | & | & | & & | \\ \mathbf{x_1} & \mathbf{x_2} & ... & \mathbf{x_{m-1}} \\ | & | & | & & | \end{bmatrix} \tag{9}$$

$$\mathbf{X'} = \begin{bmatrix} | & | & | & & | \\ \mathbf{x_2} & \mathbf{x_3} & ... & \mathbf{x_m} \\ | & | & | & & | \end{bmatrix} \tag{10}$$

and approximate the linear operator $A$ that models the trajectory $x(t)$ by formulating

$$\mathbf{X'} \approx \mathbf{AX} \tag{11}$$

Putting it all together we have the following definition.

**Definition 2** (DMD). *Suppose we have a dynamical system as in* (3) *and two sets of data (snapshots)* $\mathbf{X}$ *and* $\mathbf{X'}$ *as in* (9) *and* (10) *so that* $x'_k = F(x_k)$ *is the map in* (4) *corresponding to the map* (3) *for time* $\Delta t$. *DMD computes the leading eigendecom-* 145 *position of the best-fit linear operator* $\mathbf{A}$ *relating the data* $\mathbf{X'} \approx \mathbf{AX}$ *(H. Tu et al., 2014).*

Mathematically, the best-fit operator $\mathbf{A}$ is then defined as

$$\mathbf{A} = \arg\min_{\mathbf{A}} \|\mathbf{X'} - \mathbf{AX}\| = \mathbf{X'}\mathbf{X}^{\dagger} \tag{12}$$

where $\dagger$ denotes the Moore-Penrose pseudo-inverse of a matrix.

The validity of the approximation in (12) is supported by the following theorem, which relies on the concept of linear 150 consistency. Let us first introduce the definition of the linear consistency.

**Definition 3.** *Two matrices* $\mathbf{X} \in \mathbb{C}^{n \times m}$ *and* $\mathbf{X'} \in \mathbb{C}^{n \times m}$ *are said to be linearly consistent if for every vector* $\mathbf{c} \in \mathbb{C}^m$, *the condition* $\mathbf{Xc} = 0$ *implies* $\mathbf{X'c} = 0$.

**Theorem 1** (Tu et al., 2014, (H. Tu et al., 2014)). *If we define* $\mathbf{A} = \mathbf{X'}\mathbf{X}^{\dagger}$. *Then* $\mathbf{X'} = \mathbf{AX}$ *if and only if* $\mathbf{X'}$ *and* $\mathbf{X}$ *are linearly consistent.*

We conclude that by inverting the usual focus of the method—from modeling the system's average behavior to emphasizing its exceptional dynamics—and by carefully refining the algorithm accordingly, extreme events can be modeled more accurately across several common metrics. Moreover, this approach enables the extraction of spatial patterns that contribute to the reconstruction of extremes, along with insights into their temporal evolution.

Even though in this work we focus on the diagnostic application, the method could be easily used for future state predic-160 tion. This extension is planned for the future work. Diagnosing the evolution in time of extreme situations, one could detect reoccurring patterns and therefore predict the upcoming extremes.





### 2.3 DMD Algorithm

As mentioned before, the DMD algorithm produces the best-fit linear operator that relates the two snapshot matrices in time

1. Compute the singular value decomposition of $\mathbf{X}$ (defined in (9)), performing a low-rank truncation at the same time

$$\mathbf{X} \approx \tilde{U}\Sigma\tilde{V}^* \tag{13}$$

where * denotes the conjugate transpose, $\tilde{U} \in \mathbb{C}^{n \times r}, \tilde{\Sigma} \in \mathbb{C}^{r \times r}$, and $\tilde{V} \in \mathbb{C}^{m \times r}$, and $r \leq m$ denotes either the exact or the approximate rank of the data matrix $\mathbf{X}$

2. Project the full matrix $A$ onto $\tilde{U}$ and calculate $\tilde{A}$

$$\tilde{A} = \tilde{U}^* A \tilde{U} = \tilde{U}^* X' \tilde{V} \tilde{\Sigma}^{-1} \tag{14}$$

3. Compute the eigenvalues $\lambda_i$ (diagonal of matrix $\Lambda$) and eigenvectors $w_i$ (columns of $W$) of $\tilde{A}$. These eigenvalues and eigenvectors also correspond to the ones of the full matrix $A$.

$$\tilde{A}W = W\Lambda \tag{15}$$

4. The high-dimensional DMD modes $\Phi$ are reconstructed using the eigenvectors $W$ and the time-shifted snapshot matrix $X'$. These DMD modes are eigenvectors of the high-dimensional matrix $A$ corresponding to the eigenvalues in $\Lambda$.

$$A\Phi = \Phi\Lambda, \quad \Phi = X'V\Sigma^{-1}W \tag{16}$$

5. Reconstruct the original signal

$$\mathbf{X}_{dmd} = \Phi e^{\Omega t}\mathbf{b}, \quad \mathbf{b} = \Phi^\dagger x_0 \tag{17}$$

The angular frequency $\omega_k$ associated with each eigenvalue $\lambda_k$ is calculated $\omega_k = \frac{\ln(\lambda_k)}{\Delta t} b$, though alternative approaches exist. In our experiments, we will further explore these alternatives using optimization techniques.

### 2.4 Optimisation

Our idea is closely related to the sparse DMD (Rudy et al., 2016), where the aim is to reconstruct the original signal using as few modes as possible. In other words, to find the solution that exhibits the best balance of accuracy and sparsity. This is implemented by adding a regularisation term $\gamma$ in the regression step (8), i.e. the $L^1$ penalty.

$$min_{\mathbf{b}} J(\mathbf{b}) + \gamma \sum_{i=1}^{r} |b_i| \tag{18}$$





where the objective function $J(b)$ is defined by

$$J(b) := \|X - \Phi D_b V_\mu\|_F^2 = \sqrt{\sum_{i=1}^{m}\sum_{j=1}^{n}\left(X_{ij} - \Phi_{ij}D_{b_{ij}}V_{\mu_{ij}}\right)^2} \tag{19}$$

$\Phi$ denotes the matrix of DMD modes, as defined in Step 4 of the algorithm above, $D_b = diag\{\mathbf{b}\}$ has the amplitudes on the diagonal and $V_\mu$ is the Vandermonde matrix holding information about the angular frequency.

The goal is to control the amplitudes of the the the vector $\mathbf{b} = \Phi^\dagger x_o$. Vector $\mathbf{b}$ is needed to reconstruct a matrix corresponding

to the time evolution of the system. It is a "starting point" for the dynamical system analysed. The central question of the sparse DMD is: What are the best modes for a system, and how can they be identified? This is a complex, non-trivial problem. The proposed method aims to automate this process by reconstructing the data using as few DMD modes as possible. But the challenge remains— Which modes should be chosen? With our DMD variation, we refine the previous question: What are the best modes for reconstructing the **extremes** of a system, and how can they be identified? To answer this question, we

modify the objective function (19) by adding an additional regularisation term that accounts for extreme events. We will call it a penalisation term, since it penalises the reconstructions in which the extremes are poorly represented.

$$J(b) = \|X - \Phi D_b V_\mu\| = \sqrt{\sum_i\sum_j\left(X_{ij} - \Phi_{ij}D_{b_{ij}}V_{\mu_{ij}}\right)^2} + \underbrace{\sum_{i\in M}\sum_{j\in M_i}\|x_{ij} - \phi_{ij}D_{b_{ij}}V_{\mu_{ij}}\|}_{\text{penalisation term}}. \tag{20}$$

where $M$ represents a set of temporal indices of extreme events and $M_i$ is a set of spatial indices for a certain extreme event $i$. Here, the penalisation term suppresses deviations from those extreme events. In this way, the objective function is optimized

for the vector $b$, favoring the amplification of *extraordinary* modes— those responsible for detecting outliers— rather than the *average* ones that approximate average behaviour.

## 3  Experiments

In our experiments, we utilize reanalysis data, as it represents the most accurate approximation to observations currently available. Although reanalyses are model-based reconstructions, they are constrained by observational data. Variables that are

directly assimilated into the reanalysis forecast model tend to align more closely with real-world measurements.

For this study, we use the ERA5 reanalysis dataset (Hersbach and Dee, 2016), downloaded on a regular $0.25° \times 0.25°$ grid. Our spatial domain is restricted to Europe, spanning 70°N to 36°N and 9°W to 32°E, yielding a resolution of $17892 \times 31224$ grid points. The temporal coverage extends every day over 82 years, from 1940 to 2022. In the following experiments we focus on temperature anomaly in Europe. Although the method can be applied to any atmospheric variable.

The assumption is that the measurements are taken from the unknown dynamics and the research question is: *Are there any coherent patterns of extremes from the hidden dynamics that can be discovered using DMD algorithm?* To answer this question we implement the new variation of a Sparse Dynamic Mode Decomposition (SPDMD), called *extreme* DMD (defined in (20)).



## 3.1 Analysis

In our experiments we compare two DMD variations, the *normal* and the *extreme* one, that includes an additional penalisation
term. Having both reconstructions, we will analyze which of both performs better and which modes are significant in each of
the reconstructions.

As previously mentioned, the significance information is encoded in the amplitude vector $\mathbf{b}$. By comparing the two amplitude
vectors, $\mathbf{b}_{normal}$ and $\mathbf{b}_{extremes}$, we can determine which modes carry greater significance and which are less relevant when
dealing with extreme events. Once these key modes are identified, the next step is to explore whether a physical interpretation
exists. We extract the *most important mode* in this manner:

$$\text{most important mode index} = \arg\max\{\mathbf{b_{extreme}} - \mathbf{b_{normal}}\} \tag{21}$$

First let us define how we extract the persistent extreme events.

## 3.2 Persistent Extreme Event Detection and Validation

Heatwaves are prolonged periods of unusually hot weather, though their exact definitions vary across the literature (Xu et al.,
2016). Typically, heatwaves are defined based on a combination of duration and high-temperature thresholds (Xu et al., 2016;
Yin et al., 2022; Casati et al., 2013), often involving percentile-based approaches or specific indices, such as the ETCCDI
indices (Vogel et al., 2020). Studies have shown that the choice of threshold influences projected changes in heatwave char-
acteristics (Vogel et al., 2020), and this choice differs based on the regional climate (Perkins-Kirkpatrick and Gibson, 2017).
Therefore, climate extremes such as heatwaves, are not only defined by peak values, they must also persist over a specific time
period within a region. In our experiments, we define this persistence as a duration of at least three consecutive timesteps.

## 3.3 Local Maximum Detection

To identify extreme persistent events, we first detect a local maximum in a snapshot. We then compare this maximum to the
following two snapshots to check if the maximum persists. The condition is that the maximum stays within the neighborhood
(e.g. 5x5 pixels).

Let $D(t,x,y)$ represent the data value at time $t$ and spatial location $(x,y)$. $N$ represent the neighborhood size for initial
detection.

For each time step $t^*$, identify $(x^*,y^*)$ as the location of a local maximum :

$$D(t^*,x^*,y^*) = \max_{(i,j)\in N} D(t^*,x^*+i,y^*+j)$$

## 3.4 Persistence Check Over Previous Time Step

For a local maximum $(x^*,y^*)$ at time $t$, check persistence over the previous two time steps, $t-1$ and $t-2$:

$$D(t-1,x^*,y^*) = \max_{(i,j)\in N} D(t-1,x^*+i,y^*+j)$$





$$D(t-2, x^*, y^*) = \max_{(i,j) \in N} D(t-2, x^*+i, y^*+j)$$

If **Local Maximum Detection** and **Persistence Check over Previous Time Steps** are satisfied, only then $(t^*, x^*, y^*)$ is considered a persistent maximum. The procedure to find minima works analogously. Importantly, the detection of extreme events is performed in the transformed modal space, rather than in the original data domain. This approach allows us to identify and isolate the specific dynamic modes that contribute most significantly to rare or extreme behavior. As a result, it provides a more robust and interpretable basis for identifying outliers, compared to simply searching for large values in the raw data. Moreover,

because the modal representation operates in a reduced-dimensional subspace, this approach is also computationally more efficient, making it well-suited for high-dimensional systems.

### 3.5   *Extreme* Dynamic Mode Decomposition

We modify the *original* DMD by adding the penalisation term, which is then solved by a convex optimization problem that balances the reconstruction error, extreme value penalisation, and regularization term. We are minimising the objective function

defined in (20). We test the method across various ranks, but represent in the following section the results for rank 50.

### 3.6   Metrics

We use 4 different metrics to measure the fitness of the results. They are:

- **Mean square error (MSE)** measures the average squared difference between predicted values $\hat{\mathbf{y}}$ and the actual (true) values $\mathbf{y}$

$$\text{MSE} = \frac{1}{n} \sum_{i=1}^{n} (\hat{y}_i - y_i)^2$$

- **L-infinity norm** ($L_\infty$) (also called the maximum norm or supremum norm) of a vector measures the largest absolute value among its components. In other words, it captures the maximum magnitude of any single element in the vector.

$$\|\mathbf{x}\|_\infty = \max_{1 \le i \le n} |x_i|$$

- **The 10-norm** is a specific case of the Lp-norm where $p = 10$. It measures the "length" of a vector by taking the 10th root of the sum of the absolute values of the vector components raised to the 10th power. Higher-order norms (like the 10-norm) raise the absolute values to a high power before summing, which amplifies the effect of large components (i.e., outliers).

$$\|\mathbf{x}\|_{10} = \left( \sum_{i=1}^{n} |x_i|^{10} \right)^{\frac{1}{10}}$$

- **Structural Similarity Index Measure (SSIM)** is a perceptual metric usually used to measure the similarity between two images (or signals) by comparing their luminance, contrast, and structural information. Unlike traditional error metrics



(like MSE), SSIM aims to model the human visual system's sensitivity to changes in structural information, making it better suited for assessing perceived image quality.

$$\text{SSIM}(x, y) = \frac{(2\mu_x\mu_y + C_1)(2\sigma_{xy} + C_2)}{(\mu_x^2 + \mu_y^2 + C_1)(\sigma_x^2 + \sigma_y^2 + C_2)}$$

where:

– $\mu_x$ and $\mu_y$ are the mean intensities of image patches $x$ and $y$, respectively:

$$\mu_x = \frac{1}{N}\sum_{i=1}^{N} x_i, \quad \mu_y = \frac{1}{N}\sum_{i=1}^{N} y_i$$

– $\sigma_x^2$ and $\sigma_y^2$ are the variances of $x$ and $y$, representing contrast:

$$\sigma_x^2 = \frac{1}{N-1}\sum_{i=1}^{N}(x_i - \mu_x)^2, \quad \sigma_y^2 = \frac{1}{N-1}\sum_{i=1}^{N}(y_i - \mu_y)^2$$

– $\sigma_{xy}$ is the covariance between $x$ and $y$, capturing structural similarity:

$$\sigma_{xy} = \frac{1}{N-1}\sum_{i=1}^{N}(x_i - \mu_x)(y_i - \mu_y)$$

– $C_1$ and $C_2$ are small constants to stabilize the division, defined as:

$$C_1 = (K_1 L)^2, \quad C_2 = (K_2 L)^2$$

where $L$ is the dynamic range of pixel values (e.g., 255 for 8-bit images), and $K_1, K_2 \ll 1$.



## 3.7 Results

### 3.7.1 Heatwave 2003

We now examine in detail the well-known heatwave that struck France in August 2003, one of the most extreme and widely studied heat events in recent history (García-Herrera et al., 2010). This heatwave was characterized by persistently high temperatures, leading to severe impacts on human health, agriculture, and infrastructure. The spatial extent of the heatwave during its peak in early to mid-August 2003 is illustrated in Fig. 4, highlighting the affected regions and the intensity of the temperature anomalies.

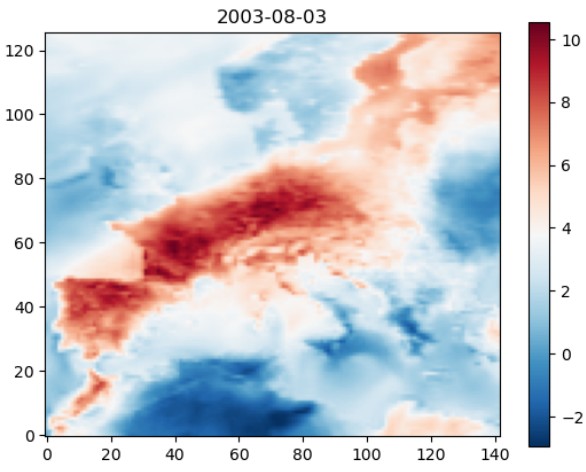

**Figure 4.** Heatwave of 2003: Red shading indicates increased temperature anomaly on August 03, 2003.

To clarify the methodology used in our experiments, in Fig. 5 we present a schematic analogue to Fig. 3, but constructed using our dataset corresponding to the 2003 heatwave. This illustration serves to provide a clearer understanding of the experimental setup and the specific data processing steps applied in our analysis.

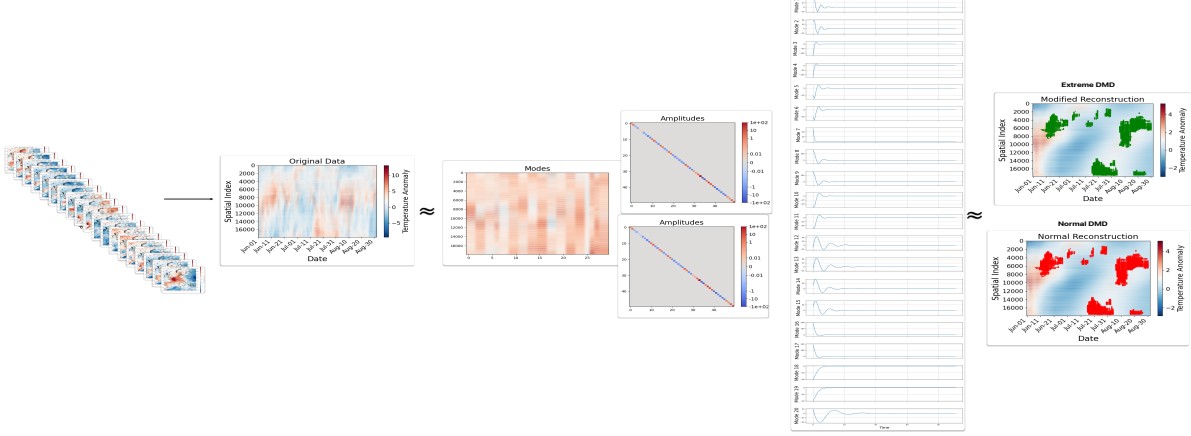

**Figure 5.** Dynamic Mode Decomposition (DMD) applied to summer 2003 data, following the same procedure as in Fig. 3. The figure compares two DMD variants in parallel: *Normal* DMD and *Extreme* DMD. The primary difference lies in the amplitudes, which lead to slightly different reconstructions of the original data.





**Which reconstruction is better?**

To show that the *extreme* DMD results indeed with better reconstruction, we show the following resulting reconstructions.

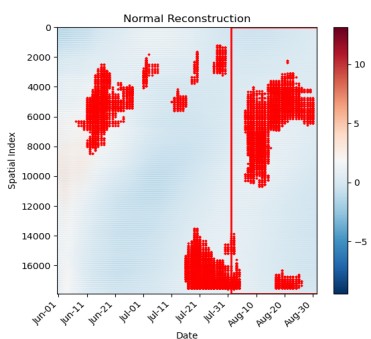 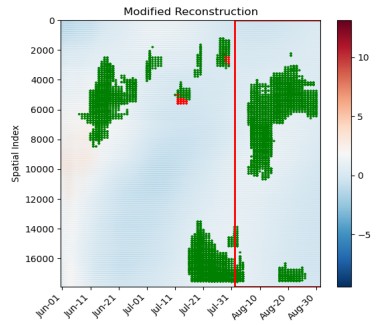 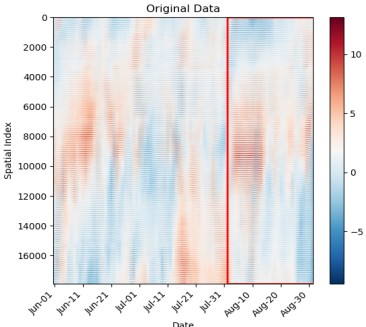

**Figure 6.** Reconstruction comparison of **Heatwave 2003.** All three plots have x-axis as a time dimension and y-axis as space dimension. They are corresponding to the flattened snapshots illustrated in the Fig. 3. The plotted markers (red and green) represent the occurrence of the extreme event. The left plot represents the reconstruction using the *normal* DMD. The middle plot represents the reconstruction using the *extreme* DMD, where green color indicates the extremes that are better reconstructed (having the smaller MSE) compared to the *normal* DMD, i.e. left plot. Whereas the right plot serves as a reference plot, representing the original values of the anomalies. Red frame indicates snapshots of only August when most of the extreme events occurred. Both plots are results of the scheme shown in Fig. 5.

In Fig. 6, the left panel illustrates the reconstruction obtained using the standard Dynamic Mode Decomposition (DMD) without a penalization term, while the middle panel presents the results of the modified *extreme* DMD approach, which incorporates a
penalization term. The rightmost panel displays the original data for reference. The dots in the plots denote extreme anomalies, with those in the middle panel highlighted in green when the reconstructed values closer match the original extreme events, as indicated by a lower mean squared error (MSE) and other metrics (see Fig. 7. and Fig. 8). We focus on August (indicated with the red rectangular in Fig 6.), the peak of the heatwave period.




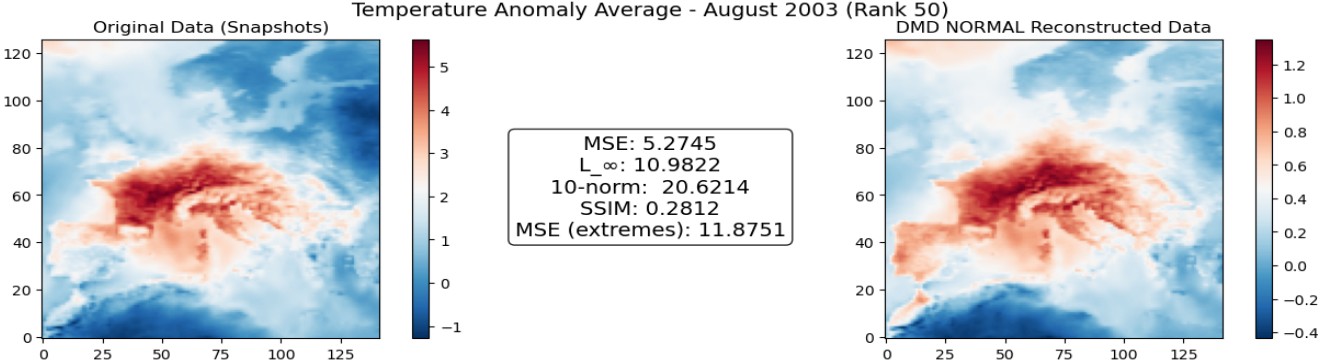

**Figure 7.** The left panel shows the average temperature anomaly in August based on the original ERA5 data (reference), while the right panel shows the corresponding reconstruction using the *normal* DMD method. The middle panel presents various metrics that evaluate the quality of the reconstruction.

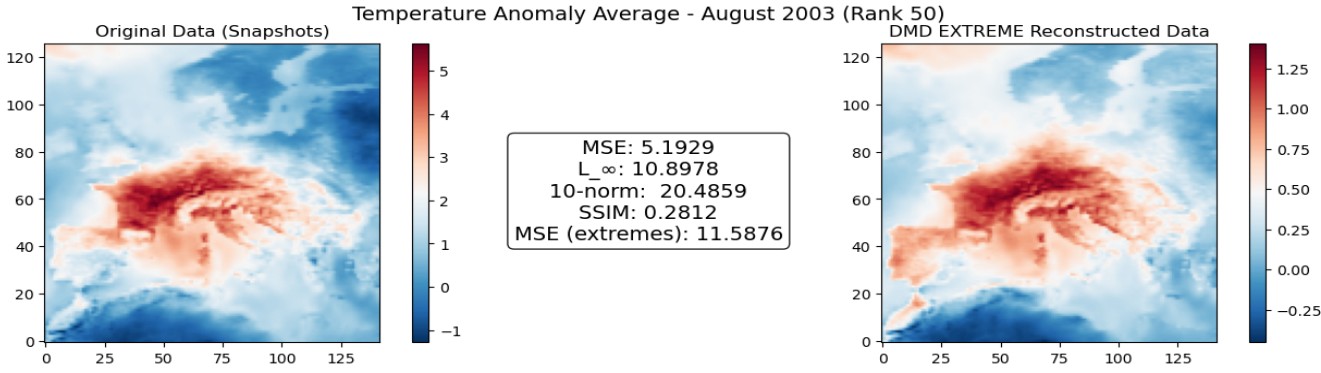

**Figure 8.** The left panel shows the average temperature anomaly in August based on the original ERA5 data (reference), while the right panel shows the corresponding reconstruction using the novel *extreme* DMD method. The middle panel presents various metrics that evaluate the quality of the reconstruction.

By analyzing and comparing various metrics presented in Fig. 7. and Fig. 8., we observe that the differences are subtle but systematic and aligned with our goal of capturing extremes. The proposed method - *extreme DMD* (results in Fig. 8) - offers a more accurate representation of extreme values.

**Which modes have more significance and which less in each of the reconstructions?**

After running two different optimisation problems: without penalising the extreme events (minimising (19)) and once with penalising the extreme events (minimising (20)), we compare the resulting optimal solutions - the amplitudes **b**'s - and we search for the biggest absolute difference.


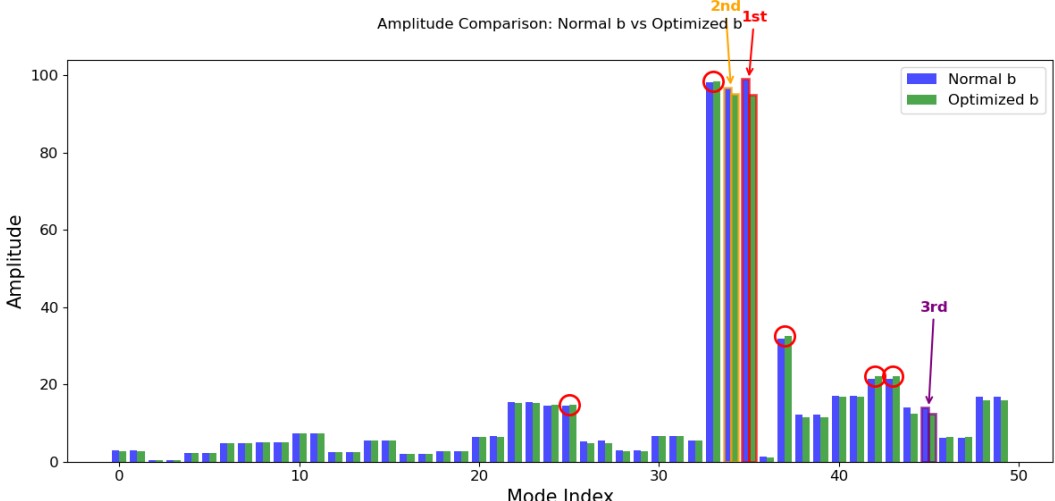

**Figure 9.** Amplitude differences between modes. The largest absolute difference is highlighted with a red frame, the second largest with an orange frame, and the third with a purple frame. Red circles indicate modes that deviate from the general trend, exhibiting higher amplitude values in the *Extreme* DMD compared to the *Normal* DMD.

In this experiment (with rank 50), the most important mode related to extreme events—defined by the highest amplitude difference—is mode 35. We extract this mode along with its temporal dynamics and present them in Fig. 10.

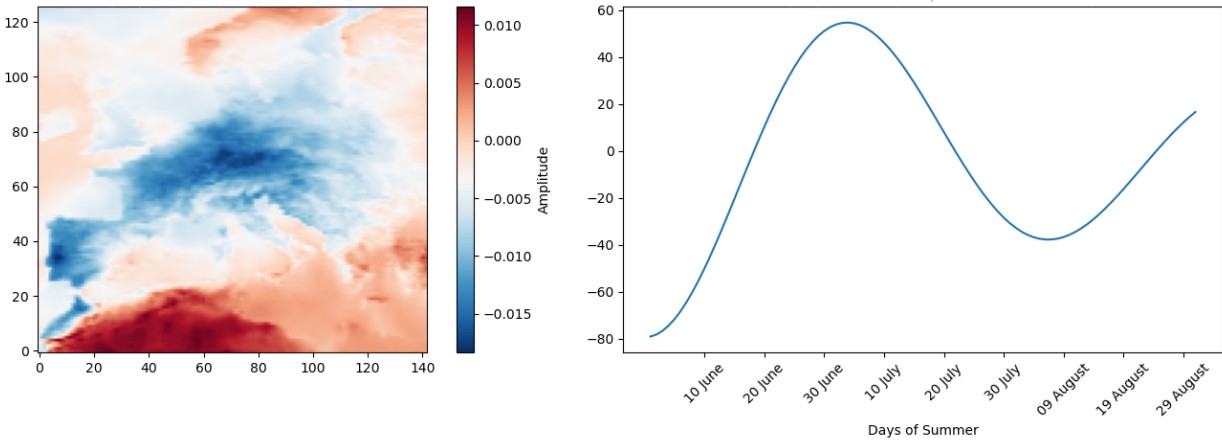

**Figure 10.** Spatial pattern and the dynamics of the *1st most important mode*.

Based on Fig. 10 illustrates that the pronounced anomaly contrast between Europe and northern Africa is more influential in shaping the average behavior of temperature anomalies than the extreme events. The temporal evolution of this mode reaches its maximum influence in late June, as evident on the right side of the panel.





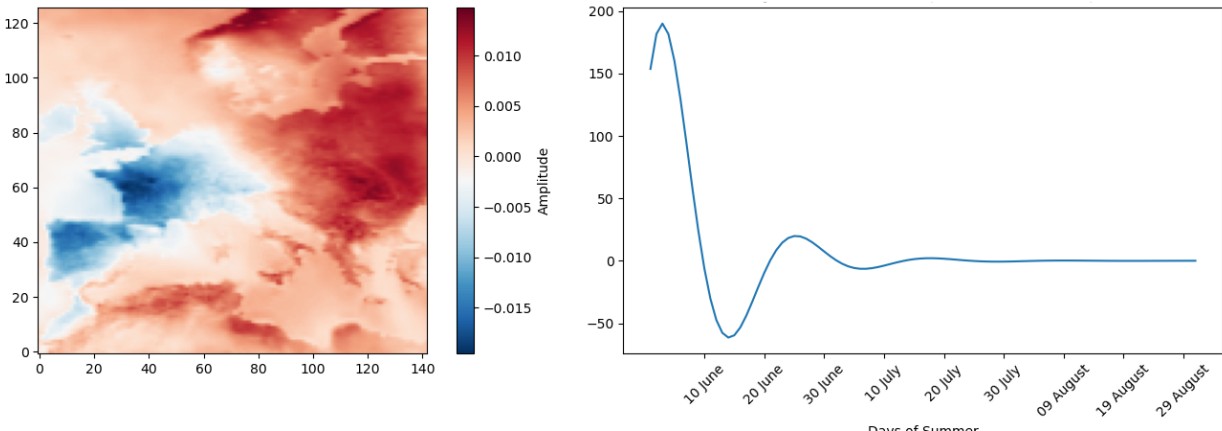

**Figure 11.** Spatial pattern and dynamics of the mode exhibiting the greatest increase in Optimized **b** ($b_{extreme}$) value relative to the Normal **b** ($b_{normal}$) value (red circles in the Fig. 9).

A spatial pattern exhibiting higher amplitude values in the extreme DMD reconstruction (unlike the others) shown in Fig. 11 suggests that the pronounced contrast in anomalies between Eastern and Western Europe played one of the key role in driving the 2003 heatwave in France.





### 3.7.2 Heatwave 2010

In Northern Germany, the heatwave in 2010 led to unusually high temperatures, with several regions experiencing prolonged periods above 35°C (Barriopedro et al., 2011). The extreme heat contributed to drought conditions, affecting agriculture and low water levels in rivers such as the Elbe and Weser. Additionally, urban areas like Hamburg and Bremen recorded significantly above-average temperatures, causing discomfort and increasing energy demand for cooling. While the heatwave in Germany was not as intense as in Russia, it still had noticeable impacts on public health and infrastructure. Here, we present the temperature anomalies that affected Northern Europe in mid-July.

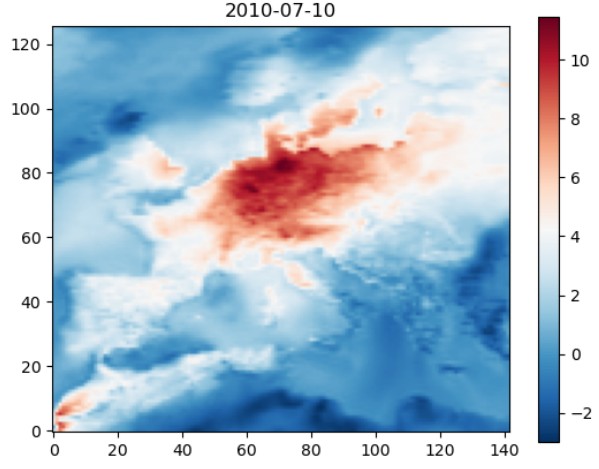

**Figure 12.** Heatwave of 2010: Red shading indicates the temperature anomaly on July 10, 2010.

**Which reconstruction is better?**

To evaluate the results, we conduct the reconstruction comparison using the same approach as in the previous experiment for the 2003 heatwave. The plotted dots denote extreme anomalies, with green highlighting regions where the reconstruction aligns more closely with the original extreme values.

The results clearly demonstrate that the *extreme* DMD approach leads to a more accurate representation of extreme temperature anomalies. This improvement is quantified by a lower mean squared error (MSE), both for the overall reconstruction and specifically for the extreme anomalies. Notably, all major extreme values observed during the 2010 heatwave are better captured using the modified DMD approach, further reinforcing the effectiveness of incorporating penalization in detecting and reconstructing extreme climate events. This behavior is further quantified by other metrics shown in Fig. 14. and Fig. 15..



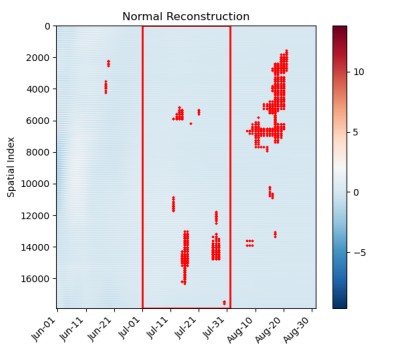 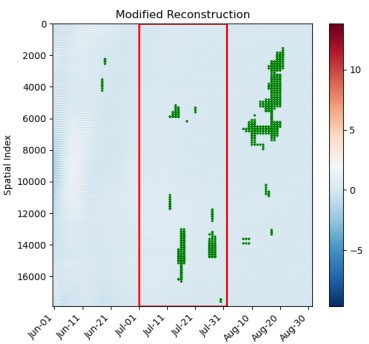 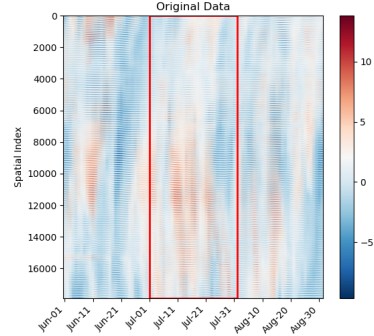

**Figure 13.** Reconstruction comparison **Heatwave 2010.** All three plots have x-axis as a time dimension and y-axis as space dimension. They are corresponding to the flattened snapshots illustrated in the Fig. 3. The plotted markers (red and green) represent the occurrence of the extreme event. The left plot represents the reconstruction using the *normal* DMD. The middle plot represents the reconstruction using the *extreme* DMD, where green color indicates the extremes that are better reconstructed (having the smaller MSE) compared to the *normal* DMD, i.e. left plot. Whereas the right plot serves as a reference plot, representing the original values of the anomalies. The red frame marks the July snapshots, corresponding to the period when extreme events had the greatest impact.

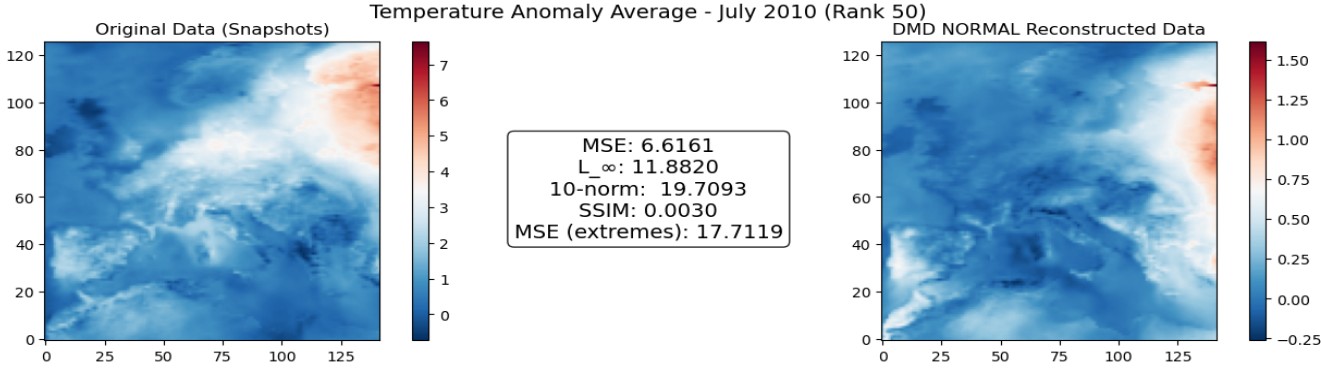

**Figure 14.** The left panel shows the average temperature anomaly in July based on the original ERA5 data (reference), while the right panel shows the corresponding reconstruction using the *normal* DMD method. The middle panel presents various metrics that evaluate the quality of the reconstruction.




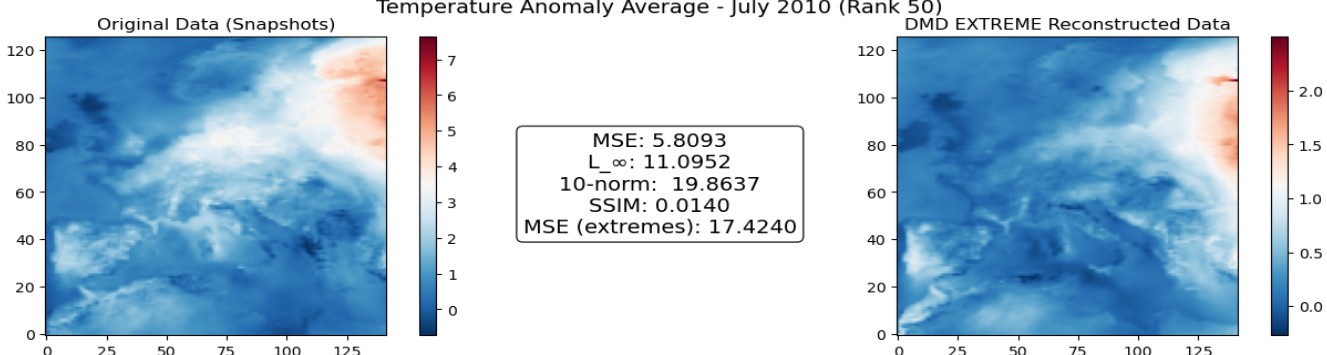

**Figure 15.** The left panel shows the average temperature anomaly in July based on the original ERA5 data (reference), while the right panel shows the corresponding reconstruction using the novel *extreme* DMD method. The middle panel presents various metrics that evaluate the quality of the reconstruction.

Consistent with earlier findings, the differences in performance metrics presented in Fig. 14. and Fig. 15. are subtle yet systematic, and once again align with our objective of enhancing the representation of extreme events. The proposed method - *Extreme DMD* (results shown in Fig. 15) - demonstrates improved accuracy in capturing extreme values.

**Which modes have more significance and which less in each of the reconstructions?**

Again we compare the amplitude vectors **b**'s which are optimisation results with the penalisation term (minimising (20)) and
without (minimising (19)).




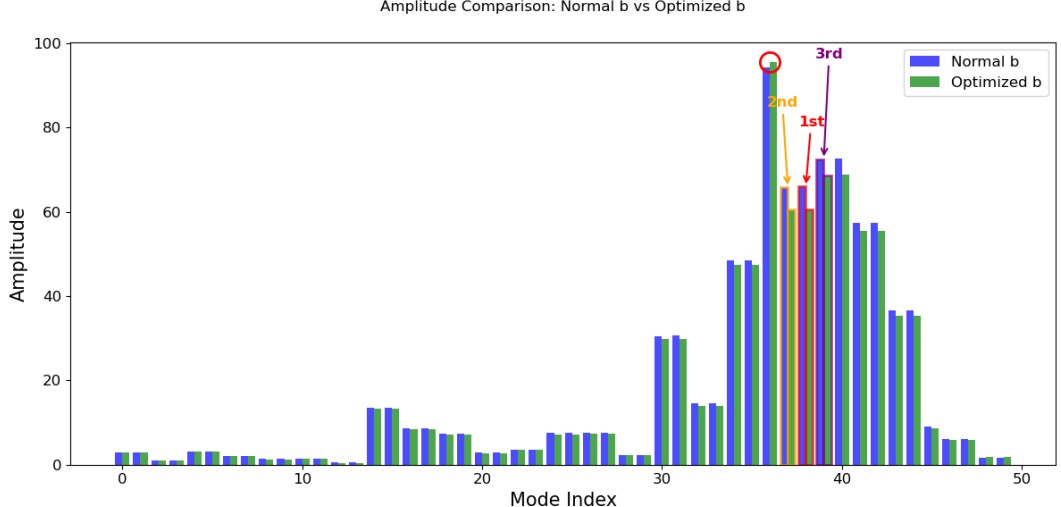

**Figure 16.** Amplitude differences between modes. The largest absolute difference is highlighted with a red frame, the second largest with an orange frame, and the third with a purple frame. Red circle indicate mode that deviates from the general trend, exhibiting higher amplitude values in the *Extreme* DMD compared to the *Normal* DMD.

The most important mode (spatial pattern) and its corresponding temporal dynamics are presented. This mode is identified based on the largest amplitude difference, as shown in Fig. 16.

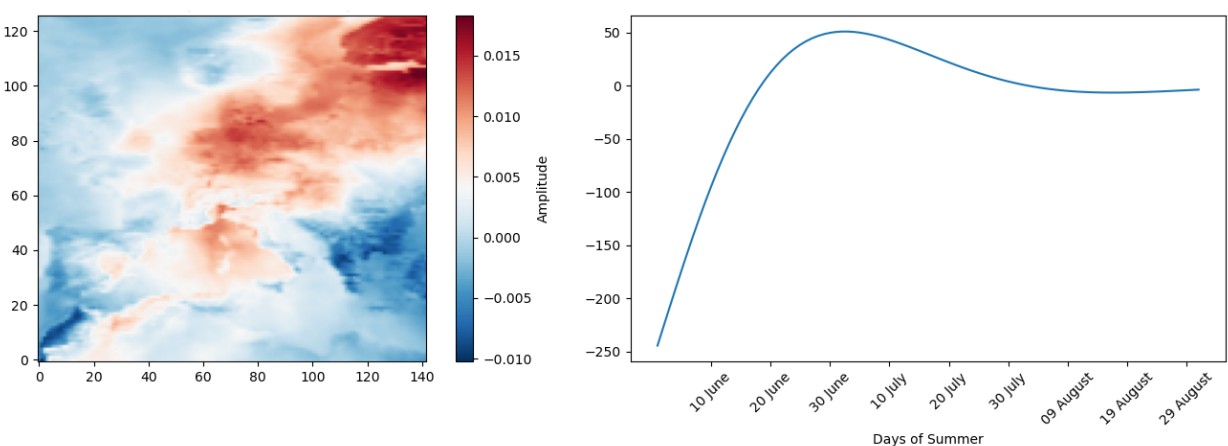

**Figure 17.** Spatial pattern and the dynamics of the *1st most important mode*.

The spatial pattern of this particular *1st most important mode* reveals the spatial pattern that strongly influences the average summer in Europe, but was less pronounced during the 2010 heatwave. It suggests a particularly strong and long-lasting 325 blocking pattern developed over Russia. This blocking event was particularly intense from early July to mid-August 2010





(Schaller et al., 2018), which is also confirmed by the dynamics showed on the right panel. The dynamics exhibited a marked increase at the beginning of summer (during June), followed by a relatively stable period with minimal oscillations throughout the remaining summer months.

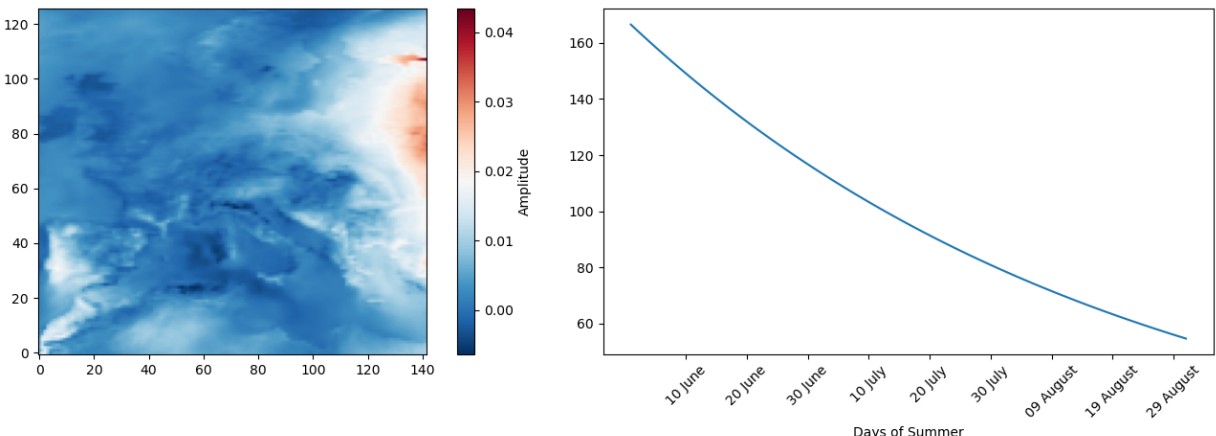

**Figure 18.** Spatial pattern and dynamics of the mode exhibiting the greatest increase in Optimized $\mathbf{b}$ ($\mathbf{b}_{extreme}$) value relative to the Normal $\mathbf{b}$ ($\mathbf{b}_{normal}$) value (red circle in the Fig. 16).

Fig. 18 shows the spatial pattern and temporal dynamics of the mode exhibiting the largest increase in Optimized $\mathbf{b}$

($\mathbf{b}_{extreme}$) compared to the Normal $\mathbf{b}$ ($\mathbf{b}_{normal}$). This mode captures a pronounced anomaly contrast between Eastern Europe (extending into Western Asia) and the rest of the continent, suggesting a strong contribution to the 2010 heatwave. The associated dynamics reveal that this spatial pattern was most prominent at the beginning of the summer and gradually weakened as the season progressed.

## 4   Discussion

Until now, reduced-order techniques have primarily focused on capturing the average behavior of a system by filtering out noise and retaining only the dominant, "typical" modes. In contrast, we challenge this perspective by asking whether the method can be reversed to reveal the exceptional modes that characterize extreme events. Hence, we introduce a theoretical framework aimed at identifying outliers together with their associated spatiotemporal modes. This allows for a deeper understanding of the underlying dynamics and helps to uncover the drivers of extreme events within the climate system. The results presented

here demonstrate that the proposed framework more effectively identifies outliers compared to the standard DMD method.

The main contribution of this work is a theoretical framework specifically designed to better detect outliers and approximate extreme - rather than average - behavior.

Our experiments clearly show that the original signal contains significantly more noise than the reconstructed fields, particularly as we move further forward in time. This is consistent with strong performance of the Koopman operator on a local





scale. However, the primary objective of these experiments was to assess the effectiveness of the proposed DMD variation in accurately capturing extreme anomalies.

One of the central concern of every reduced order technique is the decision of the number of modes needed to represent the original signal. It is always a trade off between the accuracy and the complexity. Naturally, the higher the number of modes used in the reconstruction, the better the reconstruction will be (in terms of lower mean square error). However, increasing the complexity - represented by the rank parameter in our DMD algorithm - risks introducing noise into the reconstruction while also significantly raising computational cost. Our method was tested across various ranks, with all experiments consistently showing improvements when the penalization term was added. So, we can safely claim that the method is robust with respect to the selection of the size of the rank. Unlike most of the other statistical approaches, we use rank as the only parameter, which allows for better interpretation of the model and more effective extraction of spatiotemporal patterns related to climate extremes.

Given the current excitement surrounding large language models (LLMs), it is worth to mention that LLM architecture is equivalent to the Koopman operator-based architecture. However Koopman Operator Theory offers a robust framework for unsupervised learning using small amount of data, facilitating self-supervised learning of generative models that aligns more closely with human learning theories compared to some machine learning approaches (Mezić, 2023). Furthermore, by looking at significant modes (as in Fig. 10,Fig. 11, etc.) a physical understanding of driving patterns can be derived from DMD.

The proposed framework is designed to be applicable to a broad range of climate extremes. While the current experiments have focused on temperature anomalies, specifically heatwave events, the approach can be readily extended to other types of extremes such as cold spells, heavy precipitation.

## 5 Conclusion

This work introduces a novel Dynamic Mode Decomposition (DMD) variation designed to improve the reconstruction of extreme events while providing a method to extract spatiotemporal patterns (modes) specifically relevant to such extremes. Using data-driven Koopman analysis, we demonstrate clear patterns of extremes and identify coherent behaviors in complex systems through spectral analysis of the Koopman operator. Assuming the system is well-behaved, with a diagonalizable Koopman operator and sufficient basis functions for reconstruction, we obtain a one-to-one correspondence between system dynamics and observable evolution. This approach enables us to extract interpretable coherent spatiotemporal patterns. By relying solely on time-delayed observations, this data-driven method offers insights into the system's key dynamics, serving as both a diagnostic tool and a potential prognostic model for understanding and predicting system behavior.

We conclude that by inverting the usual focus of the method—from modeling the system's average behavior to emphasizing its exceptional dynamics—and by carefully refining the algorithm accordingly, extreme events can be modeled more accurately across several common metrics. Moreover, this approach enables the extraction of spatial patterns that contribute to the reconstruction of extremes, along with insights into their temporal evolution.



Even though in this work we focus on the diagnostic application, the method could be easily used for future state prediction. This extension is planned for the future work. Diagnosing the evolution in time of extreme situations, one could detect reoccurring patterns and therefore predict upcoming extremes.

*Code availability.* The codes for generating the results are made by means of scripting Python software. All codes used in this study can be obtained from the corresponding author upon reasonable request.

*Data availability.* The ERA5 reanalysis data used in this study are publicly available from the Copernicus Climate Data Store at https://cds.climate.copernicus.eu/datasets.

*Author contributions.* MA conducted the research, performed the analyses, and wrote the initial draft of the manuscript. JB provided close
supervision throughout the project, contributed to the conceptual development, and substantially revised the manuscript. JS provided additional guidance and contributed to the manuscript revision.

*Competing interests.* The contact author has declared that none of the authors has any competing interests.

*Acknowledgements.* Authors acknowledge funding and support from the University of Hamburg. MA is grateful to the NumGeo and the Climate Extremes research group for valuable input and discussions throughout the development of this work. Additional thanks go to Dallas
Murphy for his insightful advice on scientific writing, and to the SICCS Graduate School for their support and guidance. The authors also thank the Copernicus Climate Change Service for providing the ERA5 reanalysis data. Some language editing and writing suggestions were assisted by ChatGPT (OpenAI).



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
