# Peer review of "Dynamic Mode Decomposition of Extreme Events"

_EGUsphere, 2025_

## Referee Comment (RC1)

**REVIEWER COMMENTS**

The manuscript *"Dynamic Mode Decomposition of Extreme Events"* presents a novel variation of Dynamic Mode Decomposition (DMD) by introducing a penalisation term aimed at improving the reconstruction of extreme events. The topic is timely and important, especially in the context of climate extremes, where improved diagnostics and predictive capacity are highly valuable. The manuscript is generally well-structured, but there are several issues regarding mathematical consistency, figure presentation, and the interpretation of results that need to be addressed before publication. Below, I provide general comments.

**General Comments**

**1.Clarity and Consistency of Formulas**

Some equations contain inconsistencies errors. Errors in notation can confuse readers and undermine the technical accuracy of the paper.

Line 125: The expression $\phi_j \mathbb{C}^n$ is incorrect, should be $\phi_j \in \mathbb{C}^n$.

Equation (13): The use of $\Sigma$ is inconsistent with the subsequent notation $\tilde{\Sigma}$. Please unify.

Equation (19): If the first equal sign's rhs is defined as the "square of the F-norm", then the second equal sign's rhs should not write the square root—this is mathematically inconsistent.

**2. Methodology**

In equation (20), the global residual is measured by the Frobenius norm, while the residual on the extreme set is measured by an L1 norm, and the two terms are added directly. Since the global term involves all data points while the extreme term involves only a few, their scales may not be balanced. I suggest the authors clarify whether a sensitivity analysis has been conducted, or consider introducing a weighting factor before the extreme term to ensure robustness across different datasets.

**3. Figures and Visual Presentation**

Many figures lack boundaries, latitude/longitude, clear labels, and standardized color bars, and many captions are vague, which hinders interpretation.

Fig. 9 shows multiple red circles, but Fig. 11 does not specify which is displayed.

**4.Results and Discussion**

The case study explanations of the 2003 and 2010 heatwaves are too superficial, lacking sufficient discussion of underlying physical mechanisms. Please link the extracted modes to known circulation patterns.

The method is promising, but the comparison with existing Sparse DMD or related approaches is underdeveloped. The manuscript should highlight differences and advantages of "Extreme DMD" over Sparse DMD.

**Minor Issues**

Minor typographical and formatting errors remain. A thorough proofreading is recommended. Furthermore, please ensure consistency in formatting and consider adding more recent Koopman/DMD climate applications.

**Summary and Recommendation**

Overall, the manuscript introduces an interesting and potentially valuable approach for studying climate extremes. However, issues in formula consistency, figure quality, and insufficient physical interpretation limit its current impact. I recommend **major revision** before it can be considered for publication.

---

## Referee Comment (RC2)

**Review Comments:**

This manuscript presents an extension of Dynamic Mode Decomposition (DMD) aimed at better capturing extreme climate anomalies. The proposed "extreme DMD" framework, which incorporates a penalisation term for extremes, is clearly relevant for climate science and has potential for broader applications in analyzing and predicting extreme events. The case studies on the 2003 and 2010 European heatwaves demonstrate the promise of the approach.

Below I list the specific comments:

**Major Comments**

1. The study employed two heatwave examples to demonstrate the superiority of the extreme DMD framework. Mathematical metrics (MSE, L∞, norms, SSIM) were applied to evaluate the reconstruction. However, both the comparison figures and the metrics do not reveal a clear advantage of extreme DMD over the normal DMD. In particular, for the 2003 heatwave event, the difference in MSE between the two methods is minimal, and the SSIM values are equal. Can these comparisons pass the significance test? How, then, can one convincingly demonstrate that extreme DMD possesses a stronger advantage than the normal DMD approach?

2. Since the study focuses on applying DMD to extreme events, only heatwaves are analyzed. It remains unclear whether the proposed method is applicable to other types of extremes, such as cold spells or heavy precipitation. While the authors briefly acknowledge this limitation in line 364, the issue is particularly important given that the study's title emphasizes "extremes." Without demonstrating applicability beyond heatwaves, the generality and broader relevance of the method remain uncertain. The authors should either provide additional analysis or clearly qualify the scope of their conclusions.

3. The manuscript mentions that the method could potentially be extended to prediction, but this remains unexplored. It is recommended that the authors provide a discussion on how the current results could be applied to forecasting, including potential challenges and considerations for operational implementation. It would help enhance the paper's impact on the climate community, where forecasting extreme events is a central goal.

**Minor Comments**

1. The introduction could benefit from citing more recent applications of DMD in atmospheric science, which would help better position the study within the broader climate dynamics literature.

2. Some figures (e.g., Figs. 5, 9, 16) may be difficult for readers unfamiliar with DMD to interpret. It is recommended that the authors provide additional explanations and detailed descriptions to improve clarity and accessibility.

3. Several sentences in the manuscript are lengthy and could be tightened for clarity. For instance, in Section 2.2 ("This linearisation holds only locally…"), the phrasing could be simplified to enhance readability, particularly for interdisciplinary audiences.

---

## Author Comment (AC1)

**Reviewer Responses**

November 20, 2025

*We thank all reviewers for their considerate critique and suggestions, which were valuable in revising and improving the manuscript. Detailed responses to their comments are given below.*

**Reviewer 1**

The manuscript "Dynamic Mode Decomposition of Extreme Events" presents a novel variation of Dynamic Mode Decomposition (DMD) by introducing a penalisation term aimed at improving the reconstruction of extreme events. The topic is timely and important, especially in the context of climate extremes, where improved diagnostics and predictive capacity are highly valuable. The manuscript is generally well-structured, but there are several issues regarding mathematical consistency, figure presentation, and the interpretation of results that need to be addressed before publication. Below, I provide general comments.

**General Comments**

**1. Clarity and Consistency of Formulas**

Some equations contain inconsistencies errors. Errors in notation can confuse readers and undermine the technical accuracy of the paper.

- Line 125: The expression $\phi_j \mathbb{C}^n$ is incorrect, should be $\phi_j \in \mathbb{C}^n$

  *Answer:* This expression has been corrected to $\phi_j \in \mathbb{C}^n$.

- Equation (13): The use of $\Sigma$ is inconsistent with the subsequent notation $\tilde{\Sigma}$ Please unify.

  *Answer:* It has been unified to $\tilde{\Sigma}$.

- Equation (19): If the first equal sign's rhs is defined as the "square of the F-norm", then the second equal sign's rhs should not write the square root—this is mathematically inconsistent.

  *Answer:* The equation has been corrected so that it is mathematically consistent.

**2. Methodology**

In equation (20), the global residual is measured by the Frobenius norm, while the residual on the extreme set is measured by an L1 norm, and the two terms are added directly. Since the global term involves all data points while the extreme term involves only a few, their scales may not be balanced. I suggest the authors clarify whether a sensitivity analysis has been conducted, or consider introducing a weighting factor before the extreme term to ensure robustness across different datasets.

*Answer:* We would like to clarify that both the global residual and the residual on the extreme set are measured using the Frobenius norm in our formulation. The apparent discrepancy was due to an unclear notation in equation (20) and its accompanying text. We have revised the equation and clarified in the manuscript (see lines 200–206) that the Frobenius norm is consistently used for both terms. Because both residuals use the same norm, their scales are directly comparable, and a weighting factor was not required.

**3. Figures and Visual Presentation**

Many figures lack boundaries, latitude/longitude, clear labels, and standardized color bars, and many captions are vague, which hinders interpretation. Fig. 9 shows multiple red circles, but Fig. 11 does not specify which is displayed.

*Answer:* We have revised all figures to include clear latitude - longitude boundaries, standardized color bars (where applicable), and consistent labeling. Figure captions have been rewritten to provide precise descriptions of the plotted variables, units, and the meaning of visual elements (e.g., red circles marking extreme points). In particular, the caption for Figure 11 now explicitly states which points correspond to the red circles shown in Figure 9. These updates improve visual clarity and interpretability across all figures.

**4. Results and Discussion**

The case study explanations of the 2003 and 2010 heatwaves are too superficial, lacking sufficient discussion of underlying physical mechanisms. Please link the extracted modes to known circulation patterns. The method is promising, but the comparison with existing Sparse DMD or related approaches is underdeveloped. The manuscript should highlight differences and advantages of "Extreme DMD" over Sparse DMD.

*Answer:* Our methodology is closely related to, and in part inspired by, sparse DMD. Nonetheless, it is not intended as a competing approach in the sense of being either superior or inferior. Rather, it addresses a distinct perspective. This study here serves as a "proof-of-concept" to show the merit of the "extreme DMD" as a complementary approach to sparse DMD. Real world applications (i.e. link to observed circulation patterns are beyond the scope of this study but are indeed very interesting to do in a follow up study,

**Minor Issues**

Minor typographical and formatting errors remain. A thorough proofreading is recommended. Furthermore, please ensure consistency in formatting and consider adding more recent Koopman/DMD climate applications.

*Answer:* We have carefully re-checked the manuscript and addressed typographical and formatting issues to the best of our ability, ensuring consistent formatting throughout. In addition to the previously cited Koopman/DMD climate applications at the end of the first paragraph in the introduction, we have included several more recent studies to provide an updated perspective.

**Summary and Recommendation**

Overall, the manuscript introduces an interesting and potentially valuable approach for studying climate extremes. However, issues in formula consistency, figure quality, and insufficient physical interpretation limit its current impact. I recommend major revision before it can be considered for publication.

**Reviewer 2**

This manuscript presents an extension of Dynamic Mode Decomposition (DMD) aimed at better capturing extreme climate anomalies. The proposed "extreme DMD" framework, which incorporates a penalisation term for extremes, is clearly relevant for climate science and has potential for broader applications in analyzing and predicting extreme events. The case studies on the 2003 and 2010 European heatwaves demonstrate the promise of the approach.

Below I list the specific comments:

**Major Comments**

1. The study employed two heatwave examples to demonstrate the superiority of the extreme DMD framework. Mathematical metrics (MSE, $L_\infty$ norms, SSIM) were applied to evaluate the reconstruction.

However, both the comparison figures and the metrics do not reveal a clear advantage of extreme DMD over the normal DMD. In particular, for the 2003 heatwave event, the difference in MSE between the two methods is minimal, and the SSIM values are equal. Can these comparisons pass the significance test? How, then, can one convincingly demonstrate that extreme DMD possesses a stronger advantage than the normal DMD approach?

*Answer:* We appreciate the reviewer's insightful comment and agree that a statistical assessment strengthens the comparison. To address this, we conducted formal significance testing between the Extreme DMD and Normal DMD reconstructions. For the 2003 heatwave, the MSE at extreme spatiotemporal points showed a significant improvement for the Extreme DMD (t = 90.15, p < 0.001), with a moderate-to-large effect size (Cohen's d = 0.56) and a 95% bootstrap confidence interval for the mean MSE reduction of [0.094, 0.098]. Similarly, for the 2010 heatwave, a paired t-test revealed a highly significant difference (t = 29.82, p < 0.001), with a large effect size (Cohen's d = 0.83) and a 95% bootstrap confidence interval of [0.076, 0.087]. These results, now reported in the revised manuscript, confirm that the Extreme DMD provides statistically significant and practically meaningful improvements in reconstructing extreme temperature anomalies compared to the standard DMD.

2. Since the study focuses on applying DMD to extreme events, only heatwaves are analyzed. It remains unclear whether the proposed method is applicable to other types of extremes, such as cold spells or heavy precipitation. While the authors briefly acknowledge this limitation in line 364, the issue is particularly important given that the study's title emphasizes "extremes." Without demonstrating applicability beyond heatwaves, the generality and broader relevance of the method remain uncertain. The authors should either provide additional analysis or clearly qualify the scope of their conclusions.

*Answer:* We thank the reviewer for raising this important point regarding the generality of the method. The aim of this study is to introduce a new DMD-based framework and provide a proof-of-concept demonstration using heatwaves as a representative class of extreme events. We agree that evaluating other types of extremes (e.g., cold spells or heavy precipitation) is beyond the scope of the present manuscript. To clarify this more explicitly, we have revised the Conclusion to state that the method is designed to be applicable to other types of extremes but requires further testing on additional climate variables and event types. We now clearly qualify the scope of our claims and highlight the extension to other extremes as an important direction for future work.

3. The manuscript mentions that the method could potentially be extended to prediction, but this remains unexplored. It is recommended that the authors provide a discussion on how the current results could be applied to forecasting, including potential challenges and considerations for operational implementation. It would help enhance the paper's impact on the climate community, where forecasting extreme events is a central goal.

*Answer:* We have added a discussion in the manuscript outlining how the Extreme DMD framework could be extended to short-term forecasting of extreme events. The discussion highlights that future states can be projected by evolving the identified modes and amplitudes, while noting practical challenges such as nonlinear dynamics, mode selection, and real-time data assimilation. We also emphasize the potential integration of Extreme DMD with ensemble or operational forecasting frameworks to enhance prediction and support risk assessment.

**Minor Comments**

1. The introduction could benefit from citing more recent applications of DMD in atmospheric science, which would help better position the study within the broader climate dynamics literature.

*Answer:* We have revised the introduction (lines 20–25) to include recent applications of DMD in atmospheric science, highlighting its growing use in climate and satellite data analysis

2. Some figures (e.g., Figs. 5, 9, 16) may be difficult for readers unfamiliar with DMD to interpret. It is recommended that the authors provide additional explanations and detailed descriptions to improve

clarity and accessibility.

*Answer:* To improve the clarity and accessibility of the figures, especially for readers less familiar with DMD, we have revised the captions of Figs. 5, 9, and 16 to provide more detailed explanations of what each panel represents and how to interpret the results. In addition, we have added a brief introductory paragraph before the first figure showing DMD results (now in Section 3.7) that explains how amplitude differences relate to the underlying physical interpretation—namely, that changes in amplitude reflect the varying importance of each spatial mode in reconstructing extreme versus average system behavior. These additions ensure that the figures can be understood without requiring prior familiarity with DMD.

3. Several sentences in the manuscript are lengthy and could be tightened for clarity. For instance, in Section 2.2 ("This linearisation holds only locally..."), the phrasing could be simplified to enhance readability, particularly for interdisciplinary audiences.

*Answer:* We have revised Section 2.2 to improve clarity and readability by simplifying long sentences, as we as other sections. In particular, the sentence beginning with "This linearisation holds only locally..." has been rewritten for conciseness and clarity. The revised text (lines 139-140) now reads: "This linearization holds locally because it approximates the system dynamics by finding the optimal linear relationship between nearby snapshots sampled in time."

**Reviewer 3**

The study presents an extension of Dynamic Mode Decomposition (DMD) by incorporating a regularization term specifically designed for extreme events, thereby enhancing the reconstruction accuracy of extreme climate phenomena. The proposed method successfully identifies extreme event mode that facilitate targeted analysis of extreme events, offering both scientific and practical value. However, the following aspects require further refinement to strengthen the manuscript:

**Detailed Recommendations**

- The Abstract could include quantitative comparisons (e.g., relative error reduction) of extreme event reconstruction performance between the proposed method and standard DMD.

  *Answer:* We thank the reviewer for this helpful suggestion. The Abstract has been revised to include a quantitative comparison of reconstruction performance. Specifically, we now report the relative error reduction achieved by the Extreme DMD - 0.45 - 0.85% across the two heatwave case studies—together with a note that this improvement is statistically significant. This information is now included in Lines 11–13 of the updated Abstract.

- The Conclusion section requires refinement to clearly delineate and distinguish the novel contributions from existing theoretical frameworks.We can delineate three novel contributions: a) Regularization design: integration of an extreme-event penalty term into the DMD objective function. b) Mode selection: Automated identification of extreme-relevant mode. c) Discovered climate science insight.

  *Answer:* The Conclusion section has been revised to explicitly highlight three novel contributions: (a) Regularization design: integration of an extreme-event penalty term into the DMD objective function; (b) Mode selection: automated identification of modes most relevant to extreme events; and (c) Climate science insight: application to European heatwaves revealing coherent spatiotemporal structures and the distinct impacts of anthropogenic emissions. These revisions clarify both the methodological innovations and the new scientific insights provided by our study.

- While the manuscript mentions sensitivity analysis regarding the selection of rank values, it does not present the corresponding results or their implications.

  *Answer:* In the revised manuscript, we have now performed and documented a sensitivity analysis of

the truncation rank. The results are provided in the Supplementary Material (Fig. A1). This analysis evaluates reconstruction performance across multiple rank values and shows that the Extreme DMD consistently achieves lower error in reproducing extreme spatiotemporal points than standard DMD for all tested ranks. These results indicate that the improvement is robust to the choice of rank, and we now reference this analysis in the main text.

- A comprehensive sensitivity analysis could be conducted to validate the rationality of extreme event selection criteria. Additionally, I'm curious about whether the selection ofextreme events outside the modal space would impact the results.

  *Answer:* To address this point, we conducted a sensitivity analysis using multiple percentile thresholds (90th, 95th, and 99th) to define extreme events. The results confirmed that the extracted spatial modes and their temporal evolution were highly consistent across all thresholds. This demonstrates that the Extreme DMD framework is robust to the choice of extreme-event criterion and that the key spatiotemporal structures driving extreme behavior remain stable across different definitions.

- The physical interpretation of the identified modes needs deeper exploration, particularly through integration with atmospheric dynamics principles to substantiate the selection of extreme event modes.Is it overlooked in previous studies? If not, we should add some references.

  *Answer:* We agree that establishing a stronger connection between the identified DMD modes and underlying atmospheric dynamics is essential for interpreting the results in a physical context. In the revised manuscript, we have expanded the discussion to emphasize this point and to clarify that the present study focuses primarily on methodological development and proof of concept. While a full dynamical attribution of the modes is beyond the scope of this work, we acknowledge that such an analysis - inking DMD modes to specific circulation features such as blocking or Rossby waves - represents an important direction for future research. We have also added references to previous studies that demonstrated similar data-driven decompositions in atmospheric science, focusing on mean variability and dominant climate modes. This situates our work within the broader literature while highlighting its novel focus on extremes. Corresponding text has been added to the Discussion (Section 4).

- (Optional) Improving code and data accessibility would enhance research reproducibility and academic exchange, aligning with current academic best practices.

  *Answer:* As stated in the manuscript, all codes used in this study are available from the corresponding author upon reasonable request.